# Increasing the Contents of Paddy Soil Available Nutrients and Crop Yield via Optimization of Nitrogen Management in a Wheat–Rice Rotation System

**DOI:** 10.3390/plants11172209

**Published:** 2022-08-25

**Authors:** Peng Ma, Ping Fan, Zhiyuan Yang, Yongjian Sun, Jun Ma

**Affiliations:** 1School of Life Science and Engineering, Southwest University of Science and Technology, Mianyang 621000, China; 2Rice Research Institute, Sichuan Agricultural University, Wenjiang, Chengdu 611130, China

**Keywords:** wheat–rice rotation, nitrogen fertilizer management, carbon pool, crop yield

## Abstract

To explore the impact of nitrogen (N) rate during the wheat season and N fertilizer management during the rice season on carbon and soil nutrient pools in paddy soil, a wheat–rice rotation system for 2 successive years was implemented. In the rotation system, a conventional N rate (Nc; 150 kg/ha) and a reduced N rate (Nr; 120 kg/ha) were applied in the wheat season. Based on an application rate of 150 kg/hm^2^ N in the rice season, three N management models were applied, in which the application ratio of base:tiller:panicle fertilizer was 20%:20%:60% in treatment M1, 30%:30%:40% in treatment M2, and 40%:40%:20% in treatment M3. Zero N was used as the control (M0). Experimental results indicate that, under Nc, the M2 management model during the rice season, improvements were seen in paddy soil urease, organic carbon, and annual yield relative to other conditions. The average organic matter and total N associated with the M2 rice management model and conventional N application during the wheat season were 5.13% and 4.95% higher, respectively, relative to the use of a reduced N application rate during the wheat season. Similarly, the average total carbon content and annual yields were 6.61% and 5.56% higher under the model M2 with conventional N application during the wheat season, respectively, relative to reduced N application after the two-year study period. These findings indicate that production and carbon fixation in paddy fields can be optimized through a conventional N application rate during the wheat season, and an M2 N management model during the rice season in southern China under a wheat–rice rotation system.

## 1. Introduction

Soil organic carbon (SOC) levels are a key determinant of crop productivity, as they directly influence the biological and physicochemical properties of soil and thereby regulate soil quality and sustained ecosystem viability in agricultural contexts [1,2]. As SOC exhibits a high degree of spatiotemporal variability, variations in this parameter can be challenging to detect in time scales of short or intermediated length. There is evidence to suggest that some of the earliest and most sensitive indicators of changes in SOC levels include dissolved organic carbon (DOC) and labile organic C (including microbial biomasscarbon (MBC) and easily oxidizablecorbon(EOC)), as they are highly responsive to shifts in soil management or land use strategies [3]. Variables that can influence the sequestration of SOC can include climatic conditions, and agricultural management practices including straw return and the application of fertilizer, all of which can increase corbon input or reduce C output to modulate overall SOC levels [4,5,6]. The selection of appropriate and sustainable agricultural management strategies is thus essential in order to optimize crop yields and SOC sequestration, with prior evidence supporting both the use of optimal N fertilizer application and crop straw return practices [7]. Returning crop straw to the soil has been well validated as a long-term management strategy capable of enhancing the productivity and overall quality of soil, facilitating efficient SOC accumulation and enhancements in grain yields when used in croplands [8,9,10]. Lou et al. [11], for example, found that straw residue return rates had a strong impact on topsoil C sequestration, such that crop straw return is a primary determinant of alterations in SOC levels. N fertilization can also improve SOC [12,13], with prior reports of reductions in SOC associated with N fertilizer application being better explained by shifts in soil aggregate stability and alterations in the soil organic matter fraction [14]. It is thus critical that interactions between SOC and N fertilizer be fully studied in agricultural contexts in order to select an optimal N application strategy, as the excessive use of N fertilizer can have a serious negative impact on the surrounding environment [15,16]. Further research is thus required to establish reliable approaches to reducing N fertilizer input while optimizing agricultural crop growth. Wheat–rice crop rotations are the most common cropping system in the Chengdu Plain region of China, with many field experiments having been performed to understand how straw return impacts the local ecosystems in these cropland settings [17,18]. These studies, however, have primarily evaluated shifts in crop yield, soil nutrient levels, and soil physicochemical properties, with insufficient information being available regarding the impact of N management and N rate optimization during the rice and wheat seasons, respectively, on soil labile corbon fractions and crop yield in a wheat–rice rotation system. As such, the present study was designed (1) to quantify the impact of N management optimization during the rice season and reduced N application rates during the wheat season on total organic carbon (TOC), MBC, EOC, and DOC levels in paddy soil under a wheat–rice rotation system, and (2) to evaluate optimal N management practices capable of improving soil quality and crop productivity in the context of a wheat–rice rotation system.

## 2. Materials and Methods

### 2.1. Experimental Site

This study was conducted at the Rice Research Institute farm of Sichuan Agricultural University, Wenjiang, Sichuan Province, China (30.70° N, 103.83° E) between October 2017 and early September 2019. Immediately prior to the initiation of the field experiment in 2017, the surface soil (0–20 cm depth) at the study site was found to contain 1.52 g/kg total N (Kjeldahl method, UDK-169, ITA), 23.89 mg/kg of available phosphorus (Mo–Sb colorimetry following H_2_SO_4_ and HClO_4_ digestion), 2.421% organic matter (K_2_Cr_2_O_7_-volumetric method), and 52.61 mg/kg available K (flame spectrometry following NH_4_OAc extraction), in addition to exhibiting a pH of 6.19 (as measured in a sample with a 1:2.5 soil to water ratio). Average air temperature and precipitations, as measured at a weather station near the experimental site during the previous growing season, are shown in Figure 1.

### 2.2. Experimental Design and Materials

All treatment conditions were established at the same study site using a split-plot design with three replicates per treatment from October 2017 to early September 2019. During the wheat season, two N application rates were tested including a conventional rate (Nc: 150 kg/ha) and a reduced rate (Nr: 120 kg/ha). Under a standard 150 kg/ha N application rate during the rice season, three different N management models (M1, M2, M3) were established in which base, tiller, and panicle fertilizers were applied at 2:2:6, 3:3:4, and 4:4:2 ratios, respectively. In addition, an M0 condition in which no N was applied was established as a control, and 75 P_2_O_5_ kg/ha and 150 K_2_O kg/ha were applied as basal fertilizers. The individual experimental plots were 15.75 m^2^ in area, and contained a 30 cm-wide ridge covered in a plastic film to ensure that water and nutrients were unable to penetrate adjoining plots.

The “Shumai 969” wheat variety provided by the Wheat Research Institute of Sichuan Agricultural University was used for this study. Wheat was harvested on May 8 in 2018 and 2019, and straw was cut into 5 cm pieces that were subsequently returned to the plots from which they had been collected. Wheat straw-derived N contributions to individual plots ranged from 13.08 to 19.88 kg/ha. The N, P, and K sources for this study were urea (N, 46.4%), phosphorus (P, P_2_O_5_, 12.0%), and potassium chloride (K, K_2_O, 60.0%), respectively. P and K were applied to the soil as a basal fertilizer one day prior to sowing or transplanting. During the wheat season, these N, P, and K fertilizers were applied at a 2:1:1 ratio.

The rice variety used for the present study was the common, high-yield “Fyou 498” indicia hybrid cultivar, which was sown in a seedbed on April 17 in 2018 and 2019, after which the seedlings were transplanted to the field on May 23 in 2018 and 2019. Seedlings were planted with a 0.333 m × 0.167 m spacing scheme, with one plant per hill. During the rice season, ordinary urea was applied as appropriate, and N, P, and K fertilizers were applied at a ratio of 2:1:2. Basal fertilizers during the rice season included phosphate (P_2_O_5_; 75 kg/ha) and potash (K_2_O; 150 kg/ha), which were applied one day prior to seedling transplantation. Tiller fertilizer was applied 7 days post-transplantation. Spike fertilizer, divided into flower-promoting fertilizer and flower-keeping fertilizer at a ratio of 5:5, was applied twice at the four-leaf and two-leaf stages.

### 2.3. Sampling and Measurements

After the rice was harvested, all straw was chopped and returned to the corresponding plot from which it was collected. The herbicide pretilachlor was additionally applied to these plots (1720 mL/ha) (Jiangsu Changlong Agrochemicals Co., Ltd., Taizhou, China). Herbicide was applied one time during the wheat seedling stage and the rice tillering stage. Imidacloprid (90 g/ha) (Hubei Xinhe Chemical Co., Ltd., Wuhan, China) and kasugamycin (1200 mL/ha) (Hubei Dibai Chemical Co., Ltd., Wuhan, China) were applied for pest and disease control in order to avoid yield losses.

On 5 September in 2018 and 2019, paddy soil samples (0–20 cm depth) were collected one day prior to rice harvesting. These samples were transported to the laboratory and allowed to air-dry in a cool, ventilated area. The samples were then ground, passed through a sieve, and stored in a dry location. Soil pH was measured based on the potential method using a soil–water mixture combined in a 1:2.5 ratio.

#### Organic Matter and Nutrient Content Determination

Organic matter content was determined via the K_2_Cr_2_O_7_–H_2_SO_4_ dilution thermal method, while total N content was assessed with Kjeldahl equipment, available P levels were established via the molybdenum-antimony colorimetric method, and available K content was determined through flame photometric determination. Soil TOC and soluble organic carbon were quantified using the potassium dichromate method. Soil readily oxidizable organic carbon was measured via potassium permanganate oxidation colorimetry, while soil microbial carbon was determined using chloroform fumigation, and urease activity was determined by indophenol blue colorimetry. After the wheat and rice crops had matured, each plot was harvested and threshed separately. The water content of the wheat and rice was dried and converted to allowable water contents of 14%, respectively. Annual yield = wheat yield + rice yield.

### 2.4. Statistical Analysis

Analyses of variance (ANOVA) and least significant difference (LSD) tests were used to compare data using SPSS v23 (Chinese Version v22.0.0.0) (Statistical Product and Service Solutions Inc., Chicago, IL, USA), with a significance threshold of *p* < 0.05. Figures were constructed using Origin Pro 2017 (OriginLab, Northampton, MA, USA).

## 3. Results

### 3.1. Effects of N Fertilizer Management on the Nutrient Contents of the Paddy Field

The effects of N fertilizer management on the nutrient contents of the paddy field at the maturity stage of rice under a wheat–rice rotation system are shown in Table 1. It can be seen that the year, N application rate, N fertilizer management model, and their interactions have either a significant or extremely significant effect on the nutrient contents of the soil. The nutrient contents of the soil were highest during 2019. The total N content of the soil in 2019 was 2.01% higher than that in 2018. The contents of soil organic matter and total N were reduced by 17.32% and 0.63% under the conventional N fertilization treatment and the reduced N fertilization treatment, respectively. Available K, ammonium N, and nitrate N under the conventional treatment increased by 5.71%, 5.75%, and 7.57%, respectively, compared to that of the relative reduced nitrogen application treatment. Soil organic matter, total N, ammonium N, and nitrate N all manifested a trend of M2 > M3 > M1 > M0 under the different nitrogen fertilizer operations. Total N content increased by 8.38%, 14.83%, and 19.02% under M3, M1, and M0, respectively, compared to that of M2. From the two-year experimental data, the optimal treatment of conventional N application in the wheat season and nitrogen management model M2 in the rice season increased the total N and other nutrient levels in paddy soils under the wheat–rice rotation system. This indicates that M2 management during the rice season under conventional N application in the wheat season was the most beneficial for improving the nutrient content of paddy soil.

### 3.2. Effects of N Application Rate in the Previous Growing Season and N Management in the Rice Season on Paddy Soil Urease

The effects of N application rate in the previous growing season and N management in the rice season on paddy soil urease levels are shown in Figure 2. Soil urease activity showed a decreasing trend as growth progressed, with the highest levels of soil urease content in the soil at the jointing, heading, and mature stages of rice in 2019. Soil urease content was the highest under the conventional N treatment at the jointing stage and the full heading stage; however, soil urease content was slightly higher under the reduced N application treatment at the mature stage. Soil urease content patterns under the different N fertilizer management conditions was M2 > M3 > M1 > M0 at each growth stage. Compared to M2, the urease content in M3, M1 and M0 increased by 15.12%, 20.38%, and 22.09% at the mature stage, respectively, and urease content was highest in the wheat season when combined with conventional N application and reduced N application under the M2 treatment in the rice season. In the jointing stage, full heading stage, and maturity stage of rice, the urease content under conventional N application in the wheat season coupled with the M2 rice season model was 5.65%, 7.43%, and 5.54% higher than that of reduced N application during the wheat season and the M2 operation during the rice season, respectively, indicating that conventional N application during the wheat season coupled with the M2 operation in the rice season is more conducive to increasing soil urease activity and improving the N supply to the rice rhizosphere soil environment.

### 3.3. Effects of N Management Optimization in the Rice Season and N Application Rate in the Wheat Season on the Paddy Soil Carbon Pool Contents

The effects of N fertilizer management on the content of soil carbon pool components in paddy fields under a wheat–rice rotation system are shown in Table 2. Soil carbon pool components at the maturity stage of rice were highest in 2019. The total organic carbon (TOC), easily oxidizable carbon (EOC), and carbon pool index (CPI) of the soil were highest under the conventional N application in the previous wheat season, while microbial carbon (MOC) was highest under reduced N application treatment. TOC content increased by 2.92% under conventional N application compared with that of reduced N application, and N fertilizer management exerted a significant effect on carbon pool components. The TOC and carbon pool index under different N fertilizer operations in the rice season both followed the same pattern: M3 > M2 > M1 > M0. Among them, TOC was relative to M2 and M1 under the treatment of M3. In addition, TOC increased by 5.94%, 9.84%, and 14.11% in M0 compared to that of M2, M1, and M3, respectively. Under the different N fertilizer management treatments, dissolved organic carbon (DOC) and EOC showed a pattern of M0 > M1 > M2 > M3, and MBC content showed a pattern of M3 > M1 > M2 > M0. The highest TOC content was found under the conventional N application treatment in the wheat season and the M2 rice season treatment. Under the reduced N application treatment in the wheat season, it was the M3 rice season treatment that yielded the highest TOC content, indicating that reasonable N fertilizer management under a wheat–rice rotation model can improve the carbon sequestration capacity of paddy fields and improve soil nutrient status.

### 3.4. Effects of N Management Optimization in the Rice Season and N Application Rate in the Rapeseed Season on the Crop Yield

The effects of N fertilizer management on annual crop yield under a wheat–rice rotation system are shown in Figure 3. Annual crop yield in 2019 was 2.36% lower than that of 2018. Annual crop yield under conventional N application in the wheat season was 6.21% higher compared with that of reduced N application. Under different N fertilizer operations in the rice season, annual crop yields showed a pattern of M2 > M3 > M1 > M0. Compared with that of M3, M1 and M0, the annual crop yield in M2 increased by 1.71%, 3.56%, and 34.47%, respectively. In both experimental years, annual yield was highest under the M2 rice season treatment operation under both conventional N application and reduced N application, as the nitrogen fertilizer was reduced by 40% (M2) compared with that of other treatments to increase rice yield, which ultimately increased annual crop yield. These findings indicate that conventional N application in the wheat season, coupled with a 40% (M2) shift in N fertilizer in the rice season, can increase annual crop yield.

## 4. Discussion

Greater root exudates and increased crop residues in response to mineral N fertilizer application were the dominant reasons for why N fertilizer application improved SOC [19]. Wu et al. [20] found that crop straw return to the field and reduced N application can increase SOC and active organic carbon content. The difference in soil conditions after applying N affects the rate of straw decomposition, thereby resulting in a difference in soil nutrient accumulation [21]; however, this study found that the optimization of N management in the rice season and N rate in the wheat season can increase the nutrient content, SOC and MBC contents, and soil urease content in paddy soil, especially when the M2 treatment in the rice season was combined with the conventional N rate in the wheat season in the wheat–rice rotation system. A plausible explanation for this finding may be that, under this treatment, the conditions for microorganism growth were more favorable for the efficient decomposition of straw, thus stimulating the increase in SOC and MBC after a 2-year experimental period. Certainly, climate, soil type, tillage and water regime, and straw return methods also play important roles in affecting SOC storage. Similar observations have been reported by other researchers [22,23]. It has been widely accepted that crop straw return can alter soil labile organic C fractions in the short term because of their rapid response and high sensitivity to management practices and environmental changes [24,25]. Straw is a C source for microbial activity, and supplies nucleation centers for aggregation; enhanced microbial activity induces the binding of residue and soil particles into macroaggregates, which subsequently contributes to the accumulation of labile C [26]. In this study, the M2 N management treatment in the rice season and the conventional N rate application in the wheat season led to the highest DOC and EOC contents. This finding may be due to the fact that high amounts of straw returned to the soil, and the N management strategies restrained the decomposition of the crop straw due to the blocking of gas exchange between the soil and the atmosphere, which prevented an increase in soil micro-organism activity and quantity, thereby reducing nutrient release from the crop straw into the soil in the short term [27]. In this study, there was a higher percentage of soil EOC in TOC under the M2 N management operation in the rice season and conventional N rate in the wheat season treatment, indicating a higher rate of organic matter decomposition and nutrient cycling, which will accelerate the conversion of nutrients from organic to inorganic through mineralization [28]. Liu et al. [29] demonstrated that straw return improved soil TOC and EOC in the 0–10 cm soil layer. These diverse results may be related to the influences of soil management practices, regional climate, and cropping rotation on the stability of soil labile organic C.

The productivity of crops is significantly influenced by the soil quality; thus, well-managed soil can support sustainable crop production and improve crop yields [30,31]. Our study showed that rice yield and annual yield increase rates were both higher under the M2 rice management operation and conventional wheat N rate in the wheat–rice rotation system, which is consistent with the results obtained for changes in the SOC pool. Furthermore, soil TOC, DOC, and EOC were significantly and positively correlated with annual yield, indicating that agricultural practices can increase crop productivity by improving the SOC content. It is possible that wheat straw that is returned to the field, loses a large amount of nutrients; however, when basic soil fertility is relatively high, the advance of N fertilizer can resolve the contradiction between the early straw decomposition and “N competition” in the rice plant, creating a suitable environment to strengthen soil water. The coupling effect of fertilizers increases soil microbial activity and reduces the mineralization and decomposition of soil organic matter, which is conducive to creating a favorable environment for rice growth. In this study, soil EOC and DOC were significantly and positively correlated with TOC concentrations at the 0–20 cm soil depth. Such correlations suggest that TOC is a major determinant of labile organic C fractions in soil. These results are consistent with those of authors [32] who reported similar correlations between soil TOC, labile organic C fractions (MBC, DOC, particulate organic C, EOC, and hot water-extractable C), and macroaggregate C in the 0–15 cm soil depth [32]. Therefore, improving soil organic matter content and enhancing potential rates of soil N immobilization according to optimal N management, as well as reasonable N fertilizer input, is crucial to lowering N losses in wheat–rice rotation cropping systems.

## 5. Conclusions

This study found that the base, tiller, and panicle fertilization scheme applied at the rate of 3:3:4 in the rice season under conventional N rate application in the wheat season can improve carbon fixation in paddy fields, improve soil nutrient status, and stabilize crop yield. When comprehensively considering the resource efficiency and environmental benefits, employing the base, tiller, and panicle fertilization scheme at the rate of 3:3:4 in the rice season under conventional N rates in the wheat season could be recommended to stabilize yields, and may represent an environmentally friendly strategy to improve crop yields in wheat–rice rotation systems in southern China.

## Figures and Tables

**Figure 1 plants-11-02209-f001:**
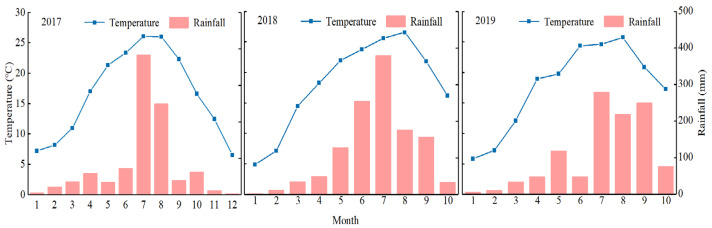
Meteorological data of the experimental area, including temperature and rain full in 2017–2019.

**Figure 2 plants-11-02209-f002:**
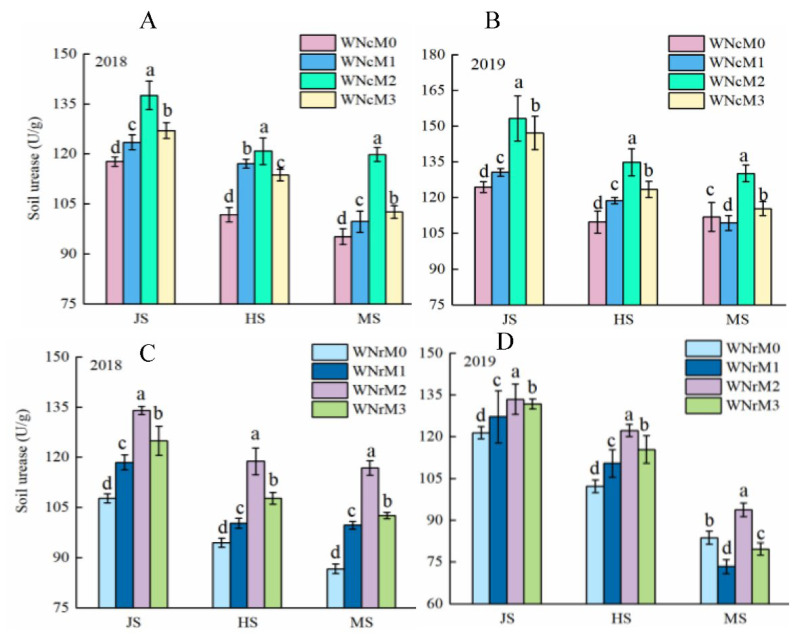
Effects of N application rate in the wheat season and N management in the rice season on paddy soil urease. JS: jointing stage; HS: heading stage; MS: maturity stage; (**A**) and (**B**) represent the conventional nitrogen application (WNc) in two years in the wheat season, (**C**) and (**D**) represent the reduced nitrogen application (WNr) in two years in the wheat season. M0 represents zero N was used in the rice season; M1, M2, and M3 represent the use of an application rate of 150 kg/hm^2^ N in the rice season, and three N management models were applied, in which the application ratio of base:tiller:panicle fertilizer was 20%:20%:60%, 30%:30%:40%, and 40%:40%:20%, respectively. Lower-case letters indicate that the soil urease levels were significantly different among the treatments (*p* < 0.05, LSD method).

**Figure 3 plants-11-02209-f003:**
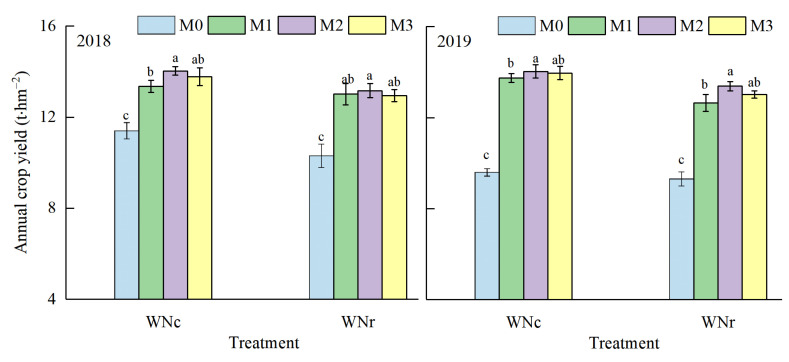
Effects of N management optimization in the rice season and N application rate in the wheat season on the crop yield. WNc and WNr represent the conventional nitrogen application and reduced nitrogen application in the wheat season, respectively. M0 represents zero N was used in rice season; M1, M2, and M3 represent the use of an application rate of 150 kg/hm^2^ N in the rice season, and three N management models were applied, in which the application ratio of base:tiller:panicle fertilizer was 20%:20%:60%, 30%:30%:40%, and 40%:40%:20%, respectively. Lower-case letters indicate that the annual crop yield was significantly different among the treatments (*p* < 0.05, LSD method).

**Table 1 plants-11-02209-t001:** The effects of N fertilizer management on the nutrient contents of the paddy field at the maturity stage.

Year	Treatment		Organic Matter (g/kg)	Total Nitrogen (g/kg)	Available Phosphorus (mg/kg)	Available Potassium (mg/kg)	Ammoniacal Nitrogen (mg/kg)	Nitrate Nitrogen (mg/kg)
**2017–2018**	Nc	M0	16.59k	1.44g	21.71o	39.29o	21.66j	10.22o
		M1	17.73hi	1.51f	31.36k	56.26h	17.92m	15.32f
		M2	25.43bc	1.68c	41.54c	57.03f	22.28i	16.17e
		M3	16.84jk	1.41g	35.37i	55.40i	35.43b	11.16m
	average		19.15	1.51	32.50	52.00	24.32	13.22
	Nr	M0	21.62e	1.28h	37.37h	49.42l	32.34e	13.23j
		M1	17.08ijk	1.41g	23.72n	37.47p	17.34n	10.33o
		M2	24.67c	1.63d	40.47e	57.47e	32.24e	17.41c
		M3	20.74f	1.51f	37.52h	53.80j	28.72g	11.77k
	average		21.03	1.46	34.77	49.54	27.66	13.19
**2018–2019**	Nc	M0	17.84hi	1.62d	26.32l	42.52m	22.46i	10.92n
		M1	18.74g	1.72bc	39.67f	59.58c	19.84k	16.98d
		M2	27.58a	1.96a	43.85a	60.17b	24.67h	17.61b
		M3	17.62hij	1.63d	33.75j	58.53d	37.44a	13.54i
	average		20.45	1.74	35.90	55.20	26.10	14.76
	Nr	M0	17.96gh	1.63d	25.49m	40.28n	18.50l	11.43l
		M1	21.82de	1.58e	41.06d	56.43g	30.74f	13.82h
		M2	25.73b	1.75b	39.01g	52.11k	34.28c	14.95g
		M3	22.58d	1.73b	43.19b	60.66a	33.43d	18.02a
	average		22.02	1.67	37.19	52.37	29.24	14.56
*F value*		Y	71.06 **	814.85 **	2872.28 **	1011.584 **	678.06 **	3954.26 **
		N	834.77 **	1.63 ns	1653.64 **	917.62 **	499.89 **	537.520 **
		M	454.85 **	245.43 **	1971.97 **	803.61 **	9578.34 **	1399.28 **
		Y × N	1.14 ns	27.24 **	120.07 **	45.18 **	9.61 *	25.84 **
		Y × M	1.07 ns	2.61 ns	504.13 **	3.18 *	13.53 **	157.64 **
		N × M	137.39 **	36.19 **	249.85 **	163.47 **	4483.36 **	2167.34 **
		Y × N × M	1.44 ns	2.94 ns	251.59 **	4.61 **	4.76 *	45.86 **

Y: year; N: nitrogen rate; M: nitrogen management; Nc and Nr represent the conventional nitrogen application and reduced nitrogen application in the wheat season, respectively. M0 represents zero N was used in rice season; M1, M2, and M3 represent the use of an application rate of 150 kg/hm^2^ N in the rice season, three N management models were applied, in which the application ratio of base:tiller:panicle fertilizer was 20%:20%:60%, 30%:30%:40%, and 40%:40%:20%, respectively. Lower-case letters indicate that the physical and chemical properties of the paddy field are significantly different among the treatments (*p* < 0.05, LSD method). * and ** mean significance at the 0.05 and 0.01 probability levels, and ns mean no significant, respectively.

**Table 2 plants-11-02209-t002:** Effects of different nitrogen treatments on soil carbon pool components in paddy fields.

Year	Treatment		TOC	DOC	MOC	EOC	CPI
			g/kg	mg/kg	mg/kg	g/kg	
**2017–2018**	Nc	M0	15.94def	138.34c	160.83p	3.34a	0.29ef
		M1	16.61cd	138.05c	196.67j	3.12bc	0.31e
		M2	17.78b	137.88c	163.07o	3.05cd	0.36cd
		M3	17.64b	136.79d	213.33f	3.01d	0.37c
	average		16.99	137.77	183.46	3.13	0.34
	Nr	M0	15.62ef	123.70g	165.85n	2.79e	0.22ij
		M1	16.09def	123.59g	286.83d	2.74e	0.25gh
		M2	16.36cde	122.33h	197.70i	2.74e	0.27fg
		M3	18.147b	114.69j	328.14a	2.60f	0.40b
	average		16.55	121.08	244.63	2.72	0.29
**2018–2019**	Nc	M0	16.64cd	140.30a	169.99m	3.40a	0.26gh
		M1	17.80b	139.12b	207.25h	3.34a	0.34d
		M2	17.67b	138.95b	173.84l	3.31a	0.36cd
		M3	17.96b	138.06c	224.24e	3.21b	0.36cd
	average		17.52	139.11	193.83	3.32	0.33
	Nr	M0	15.57f	125.76e	175.40k	3.04cd	0.21j
		M1	15.74ef	125.57e	296.78c	2.95d	0.24hi
		M2	16.90c	124.67f	208.18g	2.76e	0.29ef
		M3	19.02a	116.86i	308.62b	2.73e	0.44a
	average		16.81	123.22	247.25	2.87	0.30
*F value*		Y	11.11 **	361.304 **	5546.40 **	112.27 **	0.48 ns
		N	24.12 **	31,687.68 **	4316.56 **	696.28 **	124.54 **
		M	66.64 **	745.29 **	3037.13 **	46.18 **	303.15 **
		Y × N	1.31 ns	18.825 **	1976.77 **	1.16 ns	1.94 ns
		Y × M	0.46 ns	1.21 ns	1712.41 **	0.98 ns	4.48 *
		N × M	15.96 **	385.38 **	6661.77 **	0.97 ns	74.55 **
		Y × N × M	5.09 **	1.80 ns	1880.53 **	7.23 **	6.73 **

Y: year; N: nitrogen rate; M: nitrogen management. TOC: total organic carbon; DOC: dissolved organic carbon; MOC: microbial carbon; EOC: easily oxidizable carbon; CPI: carbon pool index. Nc and Nr represent the conventional nitrogen application and reduced nitrogen application in the wheat season, respectively. M0 represents zero N was used in rice season; M1, M2, and M3 represent the use of an application rate of 150 kg/hm^2^ N in the rice season, and three N management models were applied, in which the application ratio of base:tiller:panicle fertilizer was 20%:20%:60%, 30%:30%:40%, and 40%:40%:20%, respectively. Lower-case letters indicate that the soil carbon pool components were significantly different among the treatments (*p* < 0.05, LSD method). * and ** mean significance at the 0.05 and 0.01 probability levels, and ns mean no significant, respectively.

## Data Availability

The data presented in this study are available on request from the authors.

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
