# Peer review of "Increasing the Contents of Paddy Soil Available Nutrients and Crop Yield via Optimization of Nitrogen Management in a Wheat–Rice Rotation System"

_plants, 2022, doi:10.3390/plants11172209_

Round 1
Reviewer 1 Report
Manuscript title: Increasing the contents of paddy soil available nutrients and crop yield via optimization of nitrogen management in a wheat-rice rotation system
Manuscript ID: plants-1764171
Journal: Plants
The main objectives of the current study were to quantify the impact of N management optimization during the rice season and reduced N application rates during the wheat season on total organic carbon (TOC), MBC, EOC, and DOC levels in paddy soil under a wheat-rice rotation system. Also, the optimal N management practices capable of improving soil quality and crop productivity in the context of a wheat-rice rotation system was evaluated. The subject is interesting and the manuscript was well-written. Statistical analysis was performed correctly and well-presented in both Tables and figures. Regarding the discussion section, I suggest the authors the not divide to sub-sections and to make it more easy and readable. I also encourage the authors to reorder M&M section to be before the results even if the journal asks to this order.
Author Response
Point 1: The main objectives of the current study were to quantify the impact of N management optimization during the rice season and reduced N application rates during the wheat season on total organic carbon (TOC), MBC, EOC, and DOC levels in paddy soil under a wheat-rice rotation system. Also, the optimal N management practices capable of improving soil quality and crop productivity in the context of a wheat-rice rotation system was evaluated. The subject is interesting and the manuscript was well-written. Statistical analysis was performed correctly and well-presented in both Tables and figures. Regarding the discussion section, I suggest the authors the not divide to sub-sections and to make it more easy and readable. I also encourage the authors to reorder M&M section to be before the results even if the journal asks to this order.
Response 1: Thank you for your comment. We have already revised it(line74-151).

Reviewer 2 Report
The article presents the results of an experimental study conducted over 2 consecutive years to explore the impact of nitrogen (N) rate during the wheat season and the management of nitrogen fertilizers applied in two doses (conventional and low) and three management models in terms of the application ratio of base:tiller: panicle, during the rice season on the carbon and nutrient reservoirs in the soil from the rice soil, in the context of a wheat-rice rotation system.
The experimental design and analysis methodology are described correctly and clearly, but I don't understand why the Material and Methods Chapter is placed after the chapters of results and discussions (!?).
Chapter Results
Subchapter 2.1 Effects of N fertilizer management on the physical and chemical properties of the paddy field, refers exclusively to chemical analytical data on nutrients, without addressing the analysis of any parameters of physical characterization of the soil. It must be renamed without physical.
The experimental results are correctly interpreted and statistically analyzed, but the size of the experimental plots (15,75 mp) and the short duration of the study (only 2 years) diminish their relevance.
The experimental results conclude that the base, tiller, and panicle fertilizers applied at 3:3:4 in the rice season under conventional N rate application in the wheat season can improve carbon fixation in paddy fields, improve soil nutrient status, and stabilize crop yield in wheat-rice rotation systems in southern China.
Author Response
Response to Reviewer 2 Comments
Point1: The article presents the results of an experimental study conducted over 2 consecutive years to explore the impact of nitrogen (N) rate during the wheat season and the management of nitrogen fertilizers applied in two doses (conventional and low) and three management models in terms of the application ratio of base:tiller: panicle, during the rice season on the carbon and nutrient reservoirs in the soil from the rice soil, in the context of a wheat-rice rotation system.
The experimental design and analysis methodology are described correctly and clearly, but I don't understand why the Material and Methods Chapter is placed after the chapters of results and discussions (!?).
Response 1: Thank you for your comment. We have revised it(line74-151)
Point 2:Subchapter 2.1 Effects of N fertilizer management on the physical and chemical properties of the paddy field, refers exclusively to chemical analytical data on nutrients, without addressing the analysis of any parameters of physical characterization of the soil. It must be renamed without physical.The experimental results are correctly interpreted and statistically analyzed, but the size of the experimental plots (15,75 mp) and the short duration of the study (only 2 years) diminish their relevance.
Response 2: Thank you for your comment. We have revised it to nutrient of the paddy field(line153) ,Our test has been done in this field for many years before this study conducted in this study.

Reviewer 3 Report
Review: Plants 1764171
Title:
Increasing the contents of paddy soil available nutrients and crop yield via timization of nitrogen management in a wheat-rice rotation system
Authors: Peng Ma, Ping Fan, Zhiyuan Yang, Yongjian Sun, and Jun Ma
The aim of this review is to explore the impact of nitrogen (N) rate during the wheat season and N fertilizer man- 12 agement during rice season on carbon and soil nutrient pools in paddy soil in the context of a wheat- 13 rice rotation system, a wheat–rice rotation system for 2 successive years.
These findings indicate that production and carbon fixa- 26 tion in paddy fields can be optimized through a conventional N application rate during wheat sea- 27 son and M2 N management model during rice season in southern China under wheat-rice rotation 28 systems.
The publication presented for evaluation is very interesting.
To improve your work, please check and correct:
1. In the publication, use references to cereal phases according to the BBCH scale
2. Figure 1. It consists of 4 parts. Mark them in the order A, B, C, D. In the description of the results, refer to a specific drawing.
3. Part: Metrials and Metod – use SI units, i.e. kg/ha
4. List of literature – agree either on the full names of journals or their accepted abbreviations.
After improvement, I recommend publication for printing in the selected magazine.
Sincerely

Author Response

(The authors gave the same response as above.)

Reviewer 4 Report
In this manuscript, the author evaluated the impact of nitrogen (N) rate during the wheat season and N fertilizer management during rice season on carbon and soil nutrient pools in paddy soil. The study objective is interesting and the findings reported in this manuscript will advance the existing knowledge on N management in soils for crops. However, I have some observations as indicated below.
1. The authors obtained total N increase of 8.38%, 14.83% and 19.02% in soils with M3, M1, and M0 treatments, respectively. Surprisingly, they reported 19.02% total N increase in soils by applying zero N in Mo treatment. They also showed an increase of TOC in all the treatments as compared to Mo. I am not sure how it is possible to have this increase of total N and TOC. The authors should justify the reasons of increase of these soil parameters under inorganic N management treatments.
2. The authors maintained the allowable water content of 10.1% and 13.5% in wheat and rice grain, respectively. We usually maintain 14% water content of grain. Do they have any reference for maintaining such water content of grains?
3. In line 14, ‘a wheat-rice rotation system’ has been duplicated which needs to be corrected. Further, a space is missing in front of ‘In’ in the same line (line 14).
4. In lines 91 and 182, the authors used rapeseed season. I think it should be wheat season.
Author Response
Response to Reviewer 4 Comments
In this manuscript, the author evaluated the impact of nitrogen (N) rate during the wheat season and N fertilizer management during rice season on carbon and soil nutrient pools in paddy soil. The study objective is interesting and the findings reported in this manuscript will advance the existing knowledge on N management in soils for crops. However, I have some observations as indicated below.
Point 1: The authors obtained total N increase of 8.38%, 14.83% and 19.02% in soils with M3, M1, and M0 treatments, respectively. Surprisingly, they reported 19.02% total N increase in soils by applying zero N in Mo treatment. They also showed an increase of TOC in all the treatments as compared to Mo. I am not sure how it is possible to have this increase of total N and TOC. The authors should justify the reasons of increase of these soil parameters under inorganic N management treatments.
Response 1: Thank you for your comment. After the wheat is harvested, the straw is fully returned to field. Wheat straw decomposes can release nitrogen to increase the total organic carbon in without nitrogen field.(line105)
Point 2:The authors maintained the allowable water content of 10.1% and 13.5% in wheat and rice grain, respectively. We usually maintain 14% water content of grain. Do they have any reference for maintaining such water content of grains?
Response 2: Thank you for your comment. We have revised it to 14% water content of grain (line146)
Point 3: In line 14, ‘a wheat-rice rotation system’ has been duplicated which needs to be corrected. Further, a space is missing in front of ‘In’ in the same line (line 14).
Response 3: Thank you for your comment. We have corrected it in the revised manuscript(line14-15).
Point 4: In lines 91 and 182, the authors used rapeseed season. I think it should be wheat season.
Response 4: Thank you for your comment. We have revised it (line 172,284)
Jun Ma Peng Ma
July 20, 2022
